# IgG4-Related Chronic Sinonasal Pseudotumor with Refractory Nasal Bleeding: A Case Report

**DOI:** 10.3390/medicina58020236

**Published:** 2022-02-04

**Authors:** Hsu-Lin Lee, Meng-Ko Tsai, Deng-Ho Yang

**Affiliations:** 1Division of Hematology and Oncology, Department of Internal Medicine, Taichung Armed Forces General Hospital, Taichung 411, Taiwan; gr1027@livemail.tw; 2Division of Hematology and Oncology, Department of Internal Medicine, Tri-Service General Hospital, National Defense Medical Center, Taipei 114, Taiwan; 3Division of Rheumatology/Immunology/Allergy, Department of Internal Medicine, Taichung Armed-Forces General Hospital, Taichung 411, Taiwan; raymondpaper@gmail.com; 4Division of Rheumatology/Immunology/Allergy, Department of Internal Medicine, Tri-Service General Hospital, National Defense Medical Center, Taipei 114, Taiwan; 5Department of Medical Laboratory Science and Biotechnology, Central Taiwan University of Science and Technology, Taichung 406, Taiwan

**Keywords:** immunoglobulin G4-related disease, IgG4-RD, pseudotumor, rituximab

## Abstract

Immunoglobulin G4-related disease (IgG4-RD) is a systemic fibro-inflammatory and idiopathic autoimmune disorder. IgG4-RD can be characterized by the presence of pseudotumors. Inflammatory pseudotumors may involve any part of a human organ. There are few reports of sinus lesions in IgG4-RD. An 82-year-old man has a history of chronic sinusitis for the last several years and no remarkable family history. Two years before disease presentation, the patient experienced intermittent nasal bleeding, stuffy nose, dizziness, and fatigue. Blood test revealed positive (160X) antinuclear antibody with a mixed speckled and nucleolar pattern, IgG level of 1370 mg/dL, and IgG4 level of 99.7 mg/dL. Computed tomography (CT) of the sinus revealed several calcifications in the sphenoid sinus. Surgical findings revealed tumor-like materials. Pathological examination of the soft tissues revealed acute and chronic granulomatous inflammation. Immunohistochemical analysis demonstrated high levels of positive-affinity markers of IgG, IgG4, and CD138 and a IgG4/IgG ratio > 40%. IgG4-RD with pseudotumor was diagnosed. The initial treatment was intravenous methylprednisolone 120 mg daily for three days and oral prednisolone 10 mg twice a day and azathioprine 50 mg daily. The efficacy of the treatment was insufficient, and nasal bleeding did not decrease. Subsequently administered intravenous rituximab 1000 mg monthly for 2 months. Following this treatment, nasal bleeding stopped. CT revealed reduction in nasal mucosal swelling compared with that in a previous scan. This report highlights that in cases with an inflammatory mass mimicking malignancy, IgG4RD should always be considered, and rituximab treatment is recommended upon failure of steroid and azathioprine therapy.

## 1. Introduction

Immunoglobulin G4-related disease (IgG4-RD) is a systemic fibro-inflammatory and idiopathic autoimmune disorder. Although its pathogenesis and underlying mechanism remain unclear, IgG4-RD is characterized by the presence of pseudotumors with storiform fibrosis, obliterative phlebitis and a lymphoplasmacytic infiltrate with predominant IgG4+ plasma cells, and elevated serum IgG4 levels [1]. Inflammatory pseudotumors may involve any part of a human organ, and four distinctive phenotypes of IgG4-RD have been described according to the distribution of organ involvement [2]: group 1 (pancreato-hepatobiliary disease), group 2 (retroperitoneum and aorta), group 3 (limited to the head and neck), and group 4 (Mikulicz syndrome with systemic involvement) [2]. Clinical manifestations are progressive and locally destructive, and usually mimic malignant tumor or infectious disease [3,4]. To our knowledge, there are few reports of sinus lesions in IgG4-RD. Herein, we report a sinus pseudotumor that responded well to rituximab.

## 2. Case Report

The patient was an 82-year-old man with a history of chronic sinusitis for the last several years and no remarkable family history. Two years before disease presentation, the patient experienced intermittent nasal bleeding, stuffy nose, dizziness, and fatigue. He was treated in several hospitals and underwent functional endoscopic sinus surgery; however, the severity of these symptoms increased gradually. The patient visited our hospital in 2021. Physical examination revealed an enlarged right parotid gland and neck lymphadenopathy. No petechia or ecchymoses were observed in the body. Blood tests revealed a white blood cell count of 7900 cells/mm^3^ (reference range: 4000–10,000 cells/mm^3^), hemoglobin level of 5.3 g/dL (reference range: 13–18 g/dL), platelet count of 530,000 cells/mm^3^ (reference range: 150,000–450,000 cells/mm^3^), C-reactive protein level of 10.0 mg/dL (reference range: < 0.5 mg/dL), erythrocyte sedimentation rate of 16.0 mm/h (reference range: 2–15 mm/h), international normalized ratio of 1.03 (reference range: 0.85–1.15), activated partial thromboplastin time of 25.7 s (reference range: 23.9–35.5 s), positive(160X) antinuclear antibody with a mixed speckled and nucleolar pattern, IgG level of 1370 mg/dL (reference range: 635–1741 mg/dL), and IgG4 level of 99.7 mg/dL (reference range: 3–201 mg/dL). Computed tomography (CT) of the sinus revealed diffuse mucoperiosteal thickening in the bilateral maxillary, ethmoid, and sphenoid sinuses (Figure 1b). Several calcifications were also noted in the sphenoid sinus, suggesting fungal infection (Figure 1a). Based on these findings, we performed bilateral Caldwell Luc operations and sphenoidectomy, which revealed pus, debris, and tumor-like materials (Figure 2). The fungus culture yielded no fungal pathogen growth 2 months later. Pathological examination of the soft tissues revealed acute and chronic granulomatous inflammation (Figure 3). Immunohistochemical analysis demonstrated high levels of positive-affinity markers of IgG, IgG4, and CD138, and the IgG4/IgG ratio > 40% (Figure 3). IgG4RD with pseudotumor was diagnosed. We prescribed intravenous methylprednisolone 120 mg daily for three days, oral prednisolone 10 mg twice a day, and azathioprine 50 mg daily. The patient’s general condition improved daily, and he was discharged in stable condition at 7 days after his first visit to our hospital. Follow-up was continued for 2 months in our outpatient department. However, the efficacy of the treatment was insufficient, and nasal bleeding did not decrease. Subsequently, we administered intravenous rituximab 1000 mg twice within 6 months with the combination of prednisolone 10 mg twice a day and azathioprine 50 mg daily. Following this treatment, nasal bleeding stopped, and CT revealed reduction in nasal mucosal swelling compared with that in previous scan (Figure 1c).

## 3. Discussion

IgG4-RD is a systemic autoimmune disease with unknown etiology that frequently affects middle-aged men; it is characterized by involvement of a wide variety of organs and pseudotumor formation [5]. Inflammatory pseudotumor lacks a precise definition. Owing to its close morphological resemblance to IgG4-RD, it has recently been considered to fall within the IgG4-RD spectrum.

Diagnosis of IgG4-RD is based on clinicopathological findings and needs to incorporate features from patient history and clinical examination, as well as serological, radiological, and histopathological investigations [5]. However, it is difficult to discriminate between malignant and benign lesions and infectious disease. In the radiographic examination, inflammatory pseudotumors were visualized as ill-defined infiltrative lesions resembling malignant tumors. The lesions were isointense in T1-weighted MRI and hypointense in T2-weighted MRI, although this finding is not a pathognomonic characteristic of inflammatory pseudotumors [3]. Moreover, the serum IgG4 level is often increased, but this is not a reliable diagnostic criterion, as normal IgG4 levels have been reported in up to 40% of patients [6]. There are chances that IgG4RD may become a wastebasket entity merely based on IgG4 counts [4].

Thus, histopathological and immunohistochemical analyses of biopsy specimens remain cornerstones in the diagnosis of IgG4-RD [7]. Histopathological features show high sensitivity and specificity, facilitating accurate diagnosis [1]. The key histological features of IgG4-RD are as follows: (1) a dense lymphoplasmacytic infiltrate, (2) fibrosis organized in a storiform pattern, (3) obliterative phlebitis, and (4) a mild to moderate eosinophil infiltrate [5]. In our case, features of IgG4-RD histology revealed typical lymphoplasmacytic infiltrate with an IgG4/IgG ratio of more than 40% (Figure 2a). However, a storiform pattern and eosinophil infiltration were not noted (Figure 2). 

No standardized treatment has been established for IgG4-RD. Treatment includes surgery, steroids, immunosuppressants, radiotherapy, or a combination of these modalities [3]. Steroid treatment is most commonly used in severe cases, and excellent clinical response has been reported. Although histopathological features are considered to partly predict the response to steroid treatment, further proof is necessary to confirm this notion [1]. To date, the dose and duration of steroid therapy for IgG4-RD remain unclear, and prednisolone at a dose of 30 to 40 mg/day is commonly initiated for remission induction [5]. Relapse rates after steroid withdrawal are high and may also be significant during steroid taper or maintenance therapy [5]. Careful long-term monitoring of patients, especially those over 60 years of age, is also necessary because adverse effects of steroids are more prominent in elderly individuals [8]. Patients with IgG4-RD are refractory to steroids or easy relapse, but B cell depletion using rituximab could be a feasible option [5]. 

Keiko Ohno et al. reported two cases of IgG4-RD with nasal manifestations, for which oral steroid administration was effective [9]. In addition, Masanobu Ueno et al. reported five cases of IgG4-RD with refractory rhinitis or sinusitis, for which steroid therapy was also effective [10]. If refractory rhinitis or sinusitis is detected, IgG4-RD should be considered, and steroid treatment is recommended. However, the clinical response to steroids was poor in our case. Mikael Ebbo et al. reported rituximab as an effective treatment for IgG4-RD, although up to 40% of rituximab-treated patients relapse after 2 years of follow-up [11]. In our case, the patient’s nasal bleeding did not improve until rituximab administration. Moreover, remarkable improvement in sinusitis was observed on CT (Figure 3). The rituximab treatment is still another choice of IgG4-RD therapy.

## 4. Conclusions

Herein, we report a rare case of IgG4RD with sinonasal pseudotumors. This report highlights that in cases with an inflammatory mass mimicking malignancy. IgG4RD should always be considered, and rituximab treatment is recommended upon failure of steroid and azathioprine therapy.

## Figures and Tables

**Figure 1 medicina-58-00236-f001:**
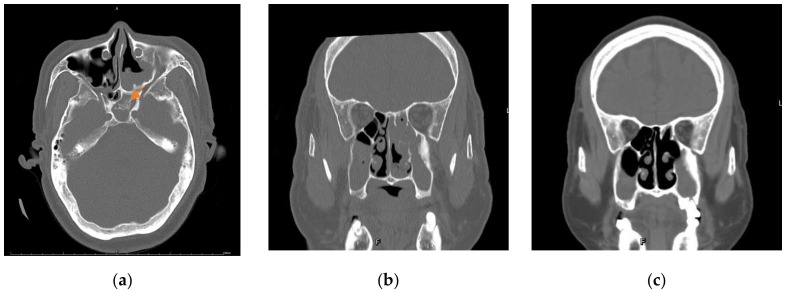
Computed tomography. (**a**) Significant calcification in the sphenoid sinus (arrow). (**b**) Diffuse mucoperiosteal thickening in the bilateral maxillary, ethmoid, and sphenoid sinuses before rituximab treatment. (**c**) Reduced inflammation in the mucosa in the bilateral maxillary, ethmoid, and sphenoid sinuses after rituximab treatment.

**Figure 2 medicina-58-00236-f002:**
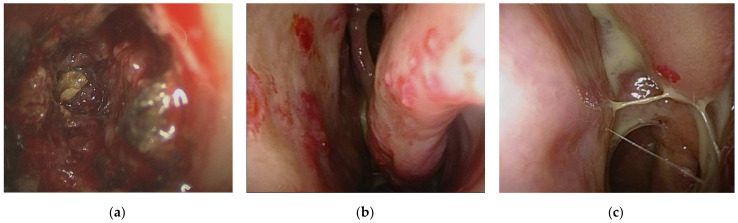
Caldwell Luc operation and sphenoidectomy. (**a**) Discharge, debris, and tumor like materials in sphenoid sinus. (**b**) Mucosal petechiae and left middle nasal conchae. (**c**) Left inferior nasal conchae.

**Figure 3 medicina-58-00236-f003:**
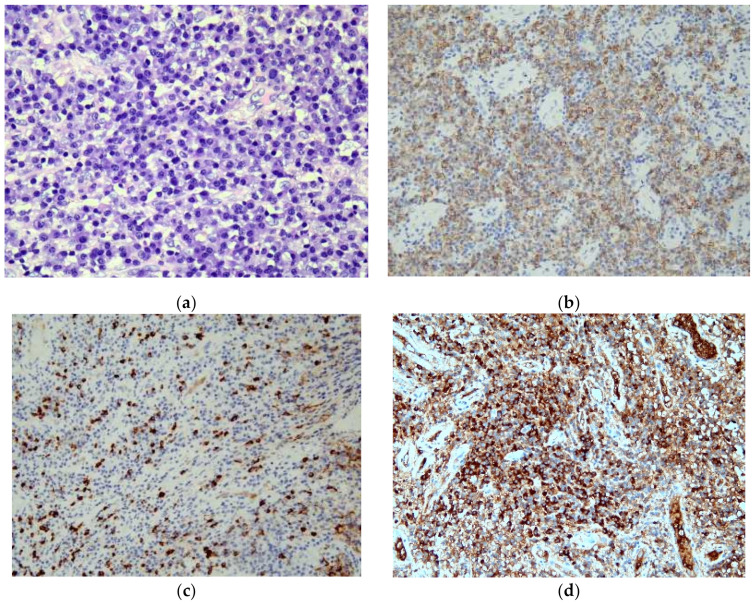
Microscopic evaluation of the lesion. (**a**) Lymphoplasmacytic infiltration observed using hematoxylin and eosin staining (magnification, 200×). (**b**) Immunohistochemistry (IHC) results for CD138 showing positivity for plasma cells (magnification, 100×). (**c**) IHC staining for IgG4 showing positivity for plasma cells (magnification, 100×). (**d**) IHC staining for IgG showing positivity for plasma cells (magnification, 100×).

## Data Availability

The data that support the findings of this study are available from the corresponding author, D.-H.Y., upon reasonable request.

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
