# Peer review of "IgG4-Related Chronic Sinonasal Pseudotumor with Refractory Nasal Bleeding: A Case Report"

_medicina, 2022, doi:10.3390/medicina58020236_

Round 1

Reviewer 1 Report

This is a single-case report on IgG4-related disease presenting with nosebleeding due to an endonasal pseudotumoral mass.

First, English language requires revision by a native speaker throughout the manuscript (check for typos, repetitions -abstract-, and grammatic).

In the abstract and conclusions (line 38 and line 141) the authors should specify that Rituximab was introduced after failure of steroid AND immunosuppressant treatment (Azathioprine).

Introduction, line 53. As far as I understood, the patient did not respond well to steroids. Please correct this statement.

Case report. CT images provided are insufficient. The authors should provide also coronal scans showing maxillary, ethmoid and sphenoid sinuses bilaterally. Please specify where did you find calcifications in the ethmoid, as I can observe it only in the left sphenoid, as I can see from the pictures provided. Did you performed also preoperative MRI for differential diagnoses with fungus ball or other conditions (benign/malignant tumor)? Please specify in the text, and if available provide MRI images.

Why did you perform Caldwell-luc procedure? Was also an endoscopic endonasal approach (FESS) taken into consideration? Was the operation intended only for diagnostic purposes (biopsy) or also to obtain the opening of the natural sinus ostia?

Please provide intraoperative images.

How long did the steroid and immunosuppressant treatment last?

How long is the follow-up? Two months (line 83) is quite short to drive any informative conclusions.

Discussion. To improve the quality of the manuscript, I suggest the authors to discuss similar cases of IgG4-RD presenting in the sinonasal tract.

Line 138: I think that “sinusoidal” is not the most appropriate term. Please change it into “sinonasal”.

Reviewer 2 Report

Dear Authors,

This case report is interesting and I think it’s worth to present the efficacy of rituximab treatment in therapy refracter IgG4-related disease.

I have some proposal for improvement the manuscript:

Line 62-67 please specify reference ranges

Figure 1 Please describe the CT features/abnormalities on each parts (a, b, c) of the figure or select only one slide (b)

Figure 2 Please specify magnification at all pictures (a, b, c, d)

Figure 3 It would be more logical to change the 2 pictures (before and after conditions). „Before condition” is the first.

Line 108 please correct „havd”

Line 132 rituximab-treated

Thanks the opportunity to review the manuscript.

Round 2

Reviewer 1 Report

Manuscript improved.